# Evaluation of the Predictive Ability, Environmental Regulation and Pharmacogenetics Utility of a BMI-Predisposing Genetic Risk Score during Childhood and Puberty

**DOI:** 10.3390/jcm9061705

**Published:** 2020-06-02

**Authors:** Augusto Anguita-Ruiz, Esther M. González-Gil, Azahara I. Rupérez, Francisco Jesús Llorente-Cantarero, Belén Pastor-Villaescusa, Jesús Alcalá-Fdez, Luis A. Moreno, Ángel Gil, Mercedes Gil-Campos, Gloria Bueno, Rosaura Leis, Concepción M. Aguilera

**Affiliations:** 1Department of Biochemistry and Molecular Biology II, Institute of Nutrition and Food Technology “José Mataix”, Center of Biomedical Research, University of Granada, Avda. del Conocimiento s/n. Armilla, 18016 Granada, Spain; augustoanguita@ugr.es (A.A.-R.); esthergg@ugr.es (E.M.G.-G.); Belen.Pastor@med.uni-muenchen.de (B.P.-V.); agil@ugr.es (Á.G.); caguiler@ugr.es (C.M.A.); 2Instituto de Investigación Biosanitaria ibs.GRANADA, 18014 Granada, Spain; 3CIBEROBN (Physiopathology of Obesity and Nutrition Network CB12/03/30038), Institute of Health Carlos III (ISCIII), 28029 Madrid, Spain; llorentefj@yahoo.es (F.J.L.-C.); lmoreno@unizar.es (L.A.M.); mercedes_gil_campos@yahoo.es (M.G.-C.); gbuenoloz@yahoo.es (G.B.); 4Growth, Exercise, Nutrition and Development (GENUD) Research Group, Instituto Agroalimentario de Aragón (IA2), Universidad de Zaragoza, Instituto de Investigación Sanitaria de Aragón (IIS Aragón), 50009 Zaragoza, Spain; airuperez@unizar.es; 5Department of Specific Didactics, Faculty of Education, University of Córdoba, 14004 Córdoba, Spain; 6PAIDI CTS-329, Maimonides Institute of Biomedical Research of Córdoba (IMIBIC), 14004 Córdoba, Spain; 7LMU—Ludwig-Maximilians-University of Munich, Division of Metabolic and Nutritional Medicine, Dr. von Hauner Children’s Hospital, University of Munich Medical Center, 80337 Munich, Germany; 8Department of Computer Science and Artificial Intelligence, University of Granada, 18071 Granada, Spain; jalcala@decsai.ugr.es; 9Unit of Pediatric Endocrinology, Reina Sofia University Hospital, 14004 Córdoba, Spain; 10Pediatric Department, Lozano Blesa University Clinical Hospital, University of Zaragoza, 50009 Zaragoza, Spain; 11Unit of Investigation in Nutrition, Growth and Human Development of Galicia, Pediatric Department, Clinic University Hospital of Santiago, University of Santiago de Compostela, 15706 Santiago de Compostela, Spain

**Keywords:** obesity, childhood obesity, metabolic syndrome, genetics, genetic risk score, pharmacogenetics, predictive ability, gene-environment interactions, puberty, childhood, Spanish children

## Abstract

Polygenetic risk scores (pGRSs) consisting of adult body mass index (BMI) genetic variants have been widely associated with obesity in children populations. The implication of such obesity pGRSs in the development of cardio-metabolic alterations during childhood as well as their utility for the clinical prediction of pubertal obesity outcomes has been barely investigated otherwise. In the present study, we evaluated the utility of an adult BMI predisposing pGRS for the prediction and pharmacological management of obesity in Spanish children, further investigating its implication in the appearance of cardio-metabolic alterations. For that purpose, we counted on genetics data from three well-characterized children populations (composed of 574, 96 and 124 individuals), following both cross-sectional and longitudinal designs, expanding childhood and puberty. As a result, we demonstrated that the pGRS is strongly associated with childhood BMI Z-Score (B = 1.56, SE = 0.27 and *p*-value = 1.90 × 10^−8^), and that could be used as a good predictor of obesity longitudinal trajectories during puberty. On the other hand, we showed that the pGRS is not associated with cardio-metabolic comorbidities in children and that certain environmental factors interact with the genetic predisposition to the disease. Finally, according to the results derived from a weight-reduction metformin intervention in children with obesity, we discarded the utility of the pGRS as a pharmacogenetics marker of metformin response.

## 1. Introduction

Among noncommunicable common diseases, overweight and obesity in children are a public health problem that has raised concern worldwide [1]. Characterized by an expansion of the adipose tissue, childhood obesity plays an important role in the development of cardio-metabolic alterations during adulthood, further increasing morbidity and mortality [2]. The early-life identification of high-risk individuals for severe obesity or cardio-metabolic alterations during adulthood is therefore indispensable to tackle down the obesity-associated mortality. A wide range of clinical and molecular factors have proven useful for obesity prediction. Among them, genetic markers are of special importance, since they allow a risk assessment from the moment of childbirth. This, combined with the strong modulatory effects of some environmental exposures, such as diet or physical activity (PA), would allow the design of personalized lifestyle plans that effectively prevent the appearance of severe obesity and cardio-metabolic alterations later in life.

Twin studies have proven a strong heritable component of body mass index (BMI), and genome-wide association studies (GWAS) have shown that adult BMI is influenced by hundreds of common genetic variants [3]. Evidence from cross-sectional and longitudinal studies has further indicated that some of these adult loci also affect BMI in childhood and puberty [4,5,6,7,8]. For many of these BMI-associated single-nucleotide polymorphisms (SNPs), significant pleiotropic genetic effects for adult cardio-metabolic traits have also been reported [9], and there is a strong evidence of a regulatory impact of environmental factors [10,11,12].

Although initial expectations for obesity GWASs were high, the results derived after two decades of research have not met the previsions (e.g., mentioned SNPs individually account for only small proportions (1–2%) of the BMI heritability [3]). Consequently, the practice of utilizing individual SNPs to predict disease is now considered a limited approach [13] and other innovative perspectives have emerged to take advantage of available GWAS insight [14]. For example, several genomic studies have proposed to study multiple common SNPs collectively to improve the estimation of disease predisposition [15]. Based on the construction of polygenic risk scores (pGRSs), that include multiple genetic variants at the same time, these approaches have recently gathered considerable interest [16] and have proven utility to identify groups of individuals who could benefit from the knowledge of their probabilistic susceptibility to disease. In brief, a pGRS is usually calculated as a weighted sum of the number of risk alleles carried by an individual, where the risk alleles and their weights are defined by the loci and their measured effects as detected by GWAS in a particular trait [17].

The inclusion of adult-BMI SNPs under a pGRS could serve therefore as an excellent predictive, and preventive, tool for facing the obesity-associated morbidity and mortality from the early periods of life. Although some previous studies have already investigated the utility of adult-BMI pGRSs for the management of obesity in children [18,19,20,21,22], no study to date has addressed the question focusing in cardio-metabolic alterations, and never under a longitudinal design comprising the metabolically risky period of puberty. In fact, puberty has been designated as the life stage where the majority of obesity-associated cardio-metabolic derangements arise [23]. The exact mechanisms connecting puberty and metabolic alterations in obesity remain unknown otherwise [24]. Furthermore, it would be interesting to elucidate to which extent BMI is due to heritable genetic factors and lifestyle behaviors; studying how the environment modulates the genetic susceptibility to disease during childhood. Beyond prognostic utility, some pGRSs have also proven to have pharmacological utility. For example, a coronary artery disease pGRS is not only able to stratify individuals by risk for disease but also by the potential clinical benefit of statin therapy [25]. However, unlike heart disease, pGRS pharmacogenetics evidences for obesity have not yet been investigated in neither children nor adults. Considering all this, we decided to evaluate the utility of an adult BMI pGRS for the prediction and pharmacological management of obesity in children, further investigating its implication in the appearance of cardio-metabolic alterations. The study design consisted of three well-characterized children populations following both cross-sectional and longitudinal approaches. For all these analyses, we employed a pGRS based on the top 44 SNPs that have previously been associated with adult BMI in the most comprehensible GWAS performed to date [3].

## 2. Objectives

(1)To demonstrate how a pGRS can quantify inherited susceptibility to obesity and its cardio-metabolic comorbidities in children.(2)To evaluate the effects of genetic predisposition for obesity during childhood and how they evolve when entering puberty.(3)To describe the plausible modulatory role of environmental factors over inherited genetic susceptibility in children.(4)To investigate the utility of the pGRS for the pharmacological management of obesity in children.

## 3. Materials and Methods

### 3.1. Study Design

The present study design consisted of three independent children populations following cross-sectional and longitudinal approaches. A general description for each study population as well as each study design are presented in Figure 1.

#### 3.1.1. Study Population 1: Cross-Sectional Approach

In order to demonstrate how the pGRS can quantify inherited susceptibility to obesity, and its cardio-metabolic comorbidities, we counted on a cross-sectional cohort of Spanish children. This cohort was referred to as study population 1 and is based on a previously conducted case-control multicentre cross-sectional design (Figure 1A) [26,27]. Among all available participants from the previous work (*N* = 1699), current genetic analyses were performed in a subset population of 574 children (293 girls) who had good quality DNA samples. Children were recruited at three Spanish health institutions: Lozano Blesa University Clinical Hospital in Zaragoza, Santiago de Compostela University Clinical Hospital in Santiago de Compostela and Reina Sofia University Clinical Hospital in Córdoba. Obesity status was defined according to BMI by using the age- and sex-specific cut-off points proposed by Cole et al. (2000) [28]. For the present analysis, there were 256 children in the obesity group, 131 in the overweight group and 187 in the normal weight group. Inclusion criteria were European-Caucasian heritage and the absence of congenital metabolic diseases. The exclusion criteria were non-European Caucasian heritage; the presence of congenital metabolic diseases (e.g., diabetes or hyperlipidaemia); undernutrition; and the use of medication that alters blood pressure, glucose or lipid metabolism. General characteristics of the 574 participants with genetics data are presented in the Appendix A.

#### 3.1.2. Study Population 2: Longitudinal Approach

With the aim of studying the effects of the pGRS on BMI changes during the course of childhood and puberty, we also performed a longitudinal analysis using data from 96 boys and girls undergoing sexual maturation (Figure 1B) recruited in the PUBMEP project (“Puberty and metabolic risk in obese children. Epigenetic alterations and pathophysiological and diagnostic implications”) [29]. Children were allocated into five experimental groups according to their obesity and insulin resistance (IR) status before and after the onset of puberty. Pubertal stage was evaluated by clinicians in all participants according to the Tanner scale (I for prepubertal and II-V for pubertal children) [30]. All details regarding the adopted longitudinal design are illustrated in Figure 1B. Obesity status was defined according to BMI by using the age- and sex-specific cut-off points proposed by Cole et al. (2000) [28]. On the other hand, the IR status was defined by means of the homeostatic model assessment for insulin resistance (HOMA-IR) index. Since HOMA-IR strongly varies with age, sex and diseases [31], and since no reference values have been yet established in neither children nor adult populations [31,32], cut-off points were extracted from a previous well-described Spanish cohort composed of 1669 children and adolescents [27,33]. For the prepubertal stage, a single cut-off value of HOMA-IR ≥ 2.5 was considered for IR [26,33]. For the pubertal stage instead, sex information was taken into consideration and different cut-off points were adopted for IR according to the 95th HOMA-IR percentile. Extracted from 778 pubertal Spanish children, pubertal IR cut-off values were HOMA-IR ≥ 3.38 in boys and HOMA-IR ≥ 3.90 in girls. Descriptive statistics for baseline data as well as longitudinal within-group and between-group changes in analyzed variables for the 96 participating children are presented in Appendix A.

#### 3.1.3. Study Population 3: RCT Metformin Clinical Intervention

In order to test whether the constructed pGRS presents utility for the pharmacological management of obesity in children, a third obesity cohort was submitted to genetic analyses in the present work. This cohort corresponded to a previous multicentre and double blind randomized controlled trial (RCT) conducted in 124 children with obesity (Figure 1C). A complete workflow detailing the study design can be found elsewhere [34,35,36]. Briefly, 160 children with obesity were stratified according to sex and pubertal status and randomly assigned to receive either (1 g/d) metformin or placebo for 6 months after meeting the defined inclusion criteria [34,35]. All the details regarding informed consent, ethics, study protocol, sample size, intervention and participants (participant’s data collection and processing, samples codification, randomization method, double-blind condition and adverse effects assessment) were previously described [34,35]. The original study was registered by European Clinical Trials Database (EudraCT, ID: 2010-023061-21) on 14 November 2011 (URL: https://www.clinicaltrialsregister.eu/ctr-search/trial/2010-023061-21/ES). Among the 160 subjects participating in the original RCT, 124 (59 placebo (29 boys) and 65 treated children (32 boys)) had an appropriate DNA sample quality for the present genetic analyses. General characteristics of the selected study population at baseline and post-treatment stages are summarized in the Appendix A. For the present pharmacogenetics analysis, differential drug response was assessed via BMI Z-score reduction after the intervention.

### 3.2. Ethics Statement

All described projects were conducted in accordance with the Declaration of Helsinki (Edinburgh 2000 revised) and followed the recommendations of the Good Clinical Practice of the CEE (Document 111/3976/88 July 1990) and the legally enforced Spanish regulation, which regulates the clinical investigation of human beings (RD 223/04 about clinical trials). The Ethics Committee on Human Research of the University of Granada (ID code: 01/2017), the Ethics Committee of the Reina Sofía University Clinical Hospital of Cordoba (ID code: 260/3408), the Bioethics Committee of the University of Santiago de Compostela (ID codes: 2011/198 and 2016/522), the Ethics Committee in Clinical Research of Aragon (ID codes: 12/2010 and 22/2016) and the Ethics Committee for Biomedical Research of Andalusia on 15 January 2012 (acta 1/12) (ID code: 2010-2739) have approved all experiments and procedures. All parents or guardians provided written informed consent, and the children gave their assent.

### 3.3. DNA Extraction, Genotyping and pGRS Construction

In all participants of the present study genomic DNA was extracted from peripheral white blood cells using two kits, the Qiamp^®^ DNA Investigator Kit for coagulated samples and the Qiamp^®^ DNA Mini & Blood Mini Kit for noncoagulated samples (QIAgen Systems, Inc., Valencia, CA, USA). All extractions were purified using a DNA Clean and Concentrator kit from Zymo Research (Zymo Research, Irvine, CA, USA). Genotyping was performed by TaqMan allelic discrimination assay using the QuantStudio 12K Flex Real-Time PCR System (Thermo Fisher Scientific, Waltham, MA, USA).

A total of 56 previously BMI-associated SNPs from the largest and most comprehensive GWAS performed to date in obesity research [3] were considered for genotyping analyses. Among them, twelve SNPs were removed in our population either due to a call rate under 95% or to a deviation from Hardy Weinberg equilibrium (Appendix A). The remaining 44 SNPs were annotated and are listed in Appendix A. Raw fluorescence measures for these genetic variants were transformed into a dosage format, where each individual genotype was represented by the number of risk alleles. Next, regression coefficients (beta-estimates) of each SNP were obtained from the GIANT consortium meta-analysis for BMI (particularly from the European population with males and females combined) [3]. The weighted pGRS was finally calculated for each individual by multiplying the number of risk alleles carried for each SNP and the corresponding extracted beta-estimate (further calculating the sum over all SNPs) (1):(1)$${\mathrm{pGRS}}_{\mathrm{BMI}}=\sum_{i=1}^{44}\beta_{{\mathrm{SNP}}_{i}}\times {\mathrm{SNP}}_{i}$$

### 3.4. Phenotypic Measurements and Lifestyle Factors

In all described cohorts, body weight (kg), height (cm) and waist circumference (cm) were measured using standardized procedures, and the BMI Z-score was calculated based on the Spanish standards reference [37]. Blood pressure was measured three times by the same examiner. Biochemical marker analyses were performed for all study populations at participating hospital laboratories following internationally accepted quality control protocols, including routine measures for lipid and glucose metabolism. Quantitative insulin sensitivity check index (QUICKI) and HOMA-IR index were calculated using fasting plasma glucose and insulin values. High-sensitivity C-reactive protein (hsCRP) was determined using a particle-enhanced turbidimetric immunoassay (Dade Behring Inc., Deerfield, IL, USA). In study populations 1 and 3, adipokines, cardiovascular risk and proinflammatory biomarkers (i.e., adiponectin, leptin, resistin, tumour necrosis factor alpha (TNF-α), interleukin (IL)-6, IL-8, total plasminogen activator inhibitor-1 (PAI-1), myeloperoxidase (MPO), matrix metalloproteinase-9 (MMP-9), soluble intercellular cell adhesion molecule-1 sICAM-1 and soluble vascular cell adhesion molecule-1 (sVCAM)) were analyzed using a Luminex 200 system (Luminex Corporation, Austin, TX, USA) with human monoclonal antibodies from Millipore (EMD Millipore Corp, Billerica, MA, USA). Descriptive statistics for all measurements were conducted in each cohort separately and can be found in Appendix A.

For study population 1, environmental exposures were further assessed through an interview that focused on PA, sedentary behaviors, disease family history and familial educational level (at mother’s and father’s levels separately). Among all available environmental data, only quantitative or ordinal variables were selected for interaction analyses.

This resulted in 47 lifestyle questions described in the Appendix A. The interviews were carried out during the school time or when children attended the consulting room at the hospital, taking approximately 30 min. In the case of PA performance and sedentary habits, they were evaluated by means of a short test based on the Physical Activity Questionnaire for Older Children (PAQ-C) and HELENA questionnaire, respectively, as well as an individual interview.

### 3.5. Statistical Analysis

#### 3.5.1. General Descriptive Analysis

All continuous variables were tested for normality using the Shapiro–Wilk test and transformed when necessary by means of the natural log or the rank-based inverse normal transformation. Heteroscedasticity between experimental groups was explored by means of the Levene test. One-way Anova, Kruskal-Wallis and the Welch test were employed to assess group differences in measurements according to standard statistical assumptions. Pairwise-*t*-tests, pairwise Mann–Whitney U-tests and Dunn tests were applied conveniently as post-hoc analyses to determine which experimental groups differ from each other. Values in descriptive tables are expressed as mean and standard deviation, or median and range if not normally distributed. In the descriptive statistics of the longitudinal cohort, within-group changes from baseline to puberty in all continuous measurements were assessed by means of a paired design; employing either a paired *t*-test or a Wilcoxon signed-rank test. Between-group differences were instead assessed by conveniently applying One-way Anova, Kruskal–Wallis or Welch tests to the computed delta values (T_1_–T_0_) for each continuous measurement.

#### 3.5.2. Association between the pGRS and Obesity Outcomes and Evaluation of the pGRS Predictive Ability

In the study population 1, logistic regression models were applied in order to test whether higher genetic risk scores were observed for subjects with obesity than for normal weight controls. A logistic regression model was further applied for comparing obesity prevalence among participants presenting a high-risk genetic profile (Q2, Q3 or Q4) versus those belonging to the reference quartile (Q1). Multiple linear regressions were employed instead to investigate the relationship between continuous measurements (including BMI Z-score) and the pGRS. For these analyses, the pGRS was treated both as a continuous and discrete variable (quartiles). To determine which SNPs within the pGRS had an independent contribution in the association with BMI Z-Score, we further performed stepwise linear regression using the “step” function included in the *stats* R package. This function uses the Akaike information criterion (AIC) to select variables for a linear model. Adjusted R^2^ and model deviance (D^2^) were calculated to assess the amount of outcome variability explained by each model. In all analyses, age, gender, pubertal stage, origin, height and BMI Z-score where adjusted for as confounders when necessary. Linear regression models were evaluated by model control (investigating linearity of effects on outcome(s), consistency with a normal distribution and variance homogeneity). All residuals- vs.-fitted, normal Q-Q, scale-location and residuals- vs.-leverage plots are available upon request. A *p*-value < 0.05 was considered as statistically significant. Given the number of analyzed markers, we also considered the false discovery rate (FDR) as in Benjamini and Hochberg to correct for multiple hypothesis testing in all analyses.

The ability of the pGRS to comprehensively discriminate between normal weight and subjects with obesity was quantified (alone or in combination with other traditional risk factors) using the area under the curve (AUC) of the receiver operating characteristic curve. This plot represents the true positive rate (sensitivity) versus the false-positive rate (specificity) and is equivalent to the overall probability that the predicted risk of an individual with disease is higher than the predicted risk of an individual without disease [38,39]. Models were first constructed based on each risk factor alone and then all models reaching an AUC ≥ 0.6 were combined. For that purpose, all samples from the study population 1 with valid data for each factor were included (not restricted to the 574 children with genetics information). Only subjects with normal weight and obesity were then selected and randomly assigned to training and test subsets in which predictive models were trained and evaluated respectively. All predictive assessments were conducted using the *PredictABEL* and the *pROC* R packages.

In order to study the ability of the pGRS as a predictor of future obesity outcomes following puberty entrance (study population 2), we performed logistic regression models with the dichotomized pGRS as an independent predictor variable (1st and 2nd tertiles vs. 3rd tertil), including the longitudinal experimental group classifications from Figure 1B as dependent dummy variables (each category vs. the reference normal weight group). Tertiles, instead of quartiles, were used here due to the low sample size of the cohort. Moreover, these models included a range of confounding factors as independent variables as indicated in the Results section. In study population 2, we further applied multiple linear regressions with deltas for continuous cardio-metabolic measurements as input variables (computed as T_1_–T_0_).

All statistical analyses were performed in R environment, version 3.6.0 (R Project for Statistical Computing).

#### 3.5.3. Identification of Gene × Environment Modulatory Effects

In the study population 1, and in each pubertal stage of the study population 2 separately, linear regression models were used to estimate the effect of gene-environment interactions (pGRS × *E*) for each collected lifestyle factor (*E*) individually. In addition to the pGRS *× E* interaction term, each tested model also included covariates such as origin and puberty, in accordance with previously published recommendations [40] (2):(2)$$\mathrm{BMI}\;Z-Score=\beta_{0}+\beta_{1}\mathrm{pGRS}+\beta_{2}E+\beta_{3}\left(\mathrm{pGRS}\times E\right)+\beta_{4}Origin+\beta_{5}Tanner+\varepsilon$$

For assessing statistical significance, we focused our attention on the estimate $\beta_{3}\left(\mathrm{pGRS}\times E\right)$ (2) and, more specifically, whether this estimate significantly deviated from zero. The null hypothesis H_0_ = 0 was either accepted or rejected, depending on the outcome of a two-sided marginal student’s *t*-test, which in this case (i.e., one degree-of-freedom difference between the nested models and normal regularity conditions) is equivalent to a likelihood-ratio test of the hypothesis H_0_ = 0. *p*-values lower than the significance level α = 0.05 were considered as statistically significant after accounting for the family-wise error rate using the FDR method. Calculations were performed in R environment, version 3.6.0 (R Project for Statistical Computing) using the “lm” function included in the *stats* package.

#### 3.5.4. pGRS-Drug Interaction

Pharmacogenetics analyses of metformin response were performed using two parallel approaches in the study population 3. First, we applied a multiple linear regression to test the effect of the pGRS on BMI Z-Score responses. For that purpose, delta changes of BMI Z-Score (computed as T_0_–T_1_) were calculated and used as the dependent variable. On the other hand, we applied a linear mixed-effects (LME) model adjusted for confounders such as tanner stage and time as fixed effects and a random intercept for each patient. Test significance for the LME model was evaluated on the pGRS:Time:Group interaction term.

## 4. Results

### 4.1. The pGRS Associates with BMI Z-Score and Performs Well in the Identification of High-Risk Children

In order to investigate the general relationship between the pGRS and obesity, we merged the anthropometric baseline data available in the cross-sectional study population 1 (*N* = 574) and the metformin-RCT study population 3 (*N* = 124) (Figure 1A,C). Descriptive statistics for each study population can be found in Appendix A. In the resulting population (*N* = 698), a model adjusted by puberty and origin showed a strongly significant association between the pGRS and the BMI Z-score (B = 1.56, SE = 0.27, *t* value = 5.69 and *p*-value = 1.90 × 10^−8^) (Figure 2). This association was quantified with an increase of 1.56 Kg of weight by each additional 0.1 of the pGRS (B = 15.6, SE = 3.93, *t* value = 5.69 and *p*-value = 8.06 × 10^−5^). The amount of BMI Z-score variance explained by the full model was 14.12%, being 4.5 the percentage of variance explained by the genetic component alone. Appendix A represents the density distribution plot of the constructed pGRS by experimental condition and the Appendix A the observed obesity: overweight: normal weight counts within each quartile of the pGRS. The pGRS followed a normal distribution in the whole study sample (D = 0.03; *p*-value = 0.08 in Lilliefors test). The mean (SD) of the pGRS in the whole sample was 1.18 (0.13), being 1.15 (0.12) in normal weight children, 1.18 (0.13) in overweight children and 1.21 (0.14) in children with obesity. After excluding overweight subjects, a logistic regression model adjusted for puberty and origin revealed a stronger risk association between the pGRS and the obesity status, so that the odds of having obesity were estimated to increase 4.7 times for each additional tenth in the pGRS (*h*^2^ = 5.6%, OR = 47.36; CI 95% = [9.8,229.38]; *p*-value = 1.64 × 10^−6^). The obesity variability attributable to the genetic component under this model was estimated in 5.6%. When comparing individuals presenting the highest risk scores (Q4 and Q3) to those belonging to the first quartile (Q1), significant associations were also evidenced (OR = 3.33; CI 95% = [1.96, 5.67]; *p*-value = 9.14 × 10^−6^ and OR = 1.66; CI 95% = [1.02, 2.71]; *p*-value = 0.04 respectively) (Appendix A).

Next, we aimed to know which SNPs contribute the most to the pGRS-BMI association. For that purpose, we performed a stepwise linear regression including all 44 tested SNPs and found the genetic variants rs543874-*LOC101928778:SEC16B*, rs7138803-*BCDIN3D:FAIM2*, rs10132280-*STXBP6:NOVA1*, rs1558902-*FTO* and rs12940622-*RPTOR* to be the most determinant polymorphisms for BMI Z-Score (Appendix A). This finding was further supported by the univariate association analyses conducted between the BMI Z-score and each individual SNP (Appendix A).

To demonstrate the validity of the pGRS for the clinical prediction of obesity (alone or in combination with other traditional risk factors), logistic regression models were constructed including different combination of risk factors and further evaluated using AUC (Table 1). For each predictive model, subjects presenting valid data for assayed variables were selected from the study population 1 and divided into a training set (composed of the 75% of total available samples) and a test set (formed by the remaining 25%). Performance statistics from each trained model in the respective test set are presented in Table 1. Among all single-variable predictive models, the model including the pGRS demonstrated one of the greater predictive abilities (only surpassed by the model including parental BMI information). The joint combination of all models individually surpassing an AUC of 0.6 yielded a considerable improvement in the predictive ability (AUC = 0.81 CI 95%= [0.7–0.93]), which could be sufficient for clinical discrimination.

### 4.2. The pGRS is Associated with Longitudinal Trajectories for Obesity and IR in Children Undergoing Puberty

With the aim of studying the effects of the pGRS on obesity during the course of puberty, we also performed a longitudinal design on 96 boys and girls undergoing sexual maturation (study population 2). All details regarding the adopted longitudinal design are illustrated in Figure 1B. The 96 individuals were stratified according to two classification criteria; (1) joint longitudinal trajectories for obesity and IR, and (2) the longitudinal trajectories for obesity. The number of resulting experimental groups per classification as well as the final sample size per group are shown in Figure 1B. Longitudinal within-group and between-group changes for all analyzed biochemical variables are shown in Appendix A, according to the experimental classification 1. Changes in anthropometric variables showed a coherent behavior according to each experimental condition. In particular, for waist circumference (WC), which is a metabolic health indicator in obesity, we found significant within-group increases accompanying sexual maturation in all groups. The higher increase corresponded to group 4, in which children with obesity become IR with pubertal maturation. The metabolic health derangement observed in groups 4 and 5 for WC was also confirmed by changes in blood pressure, insulin and glucose levels, QUICKI, HOMA-IR and triglycerides.

Regarding the pGRS, findings reported in the Results Section 4.1 (merged study populations 1 and 3) were independently validated here with the longitudinal study population 2 (N = 96), using data from each time point individually. For the prepubertal stage, a multiple linear regression analysis revealed a significant association between the pGRS and the BMI Z-Score after adjusting by origin (B = 2.84, CI 95% = [0.31, 5.37]; *p*-value = 0.03). When excluding overweight individuals from analysis, the odds of obesity were quantified as 8.22 times higher in the children belonging to the 3rd tertile of the pGRS with regard to children belonging to the 1st and 2nd tertiles (CI 95% = [1.95, 34.61]; *p*-value = 0.004). For the pubertal stage, the multiple linear regression model did not find a significant association between the continuous pGRS and the BMI Z-Score after adjusting by origin and pubertal status (B = 0.9, CI 95% = [−1.57, 3.38]; *p*-value = 0.47). Instead, when excluding overweight individuals, the odds of obesity were estimated to be 5.54 times significantly higher in pubertal children belonging to the third tertile of the pGRS in comparison to those belonging to the first two tertiles (CI 95% = [1.41, 21.52]; *p*-value = 0.01).

In order to study the ability of the pGRS to predict future outcomes after puberty entrance, we next performed logistic regression models with the dichotomized pGRS as an independent predictor variable (1st and 2nd tertiles vs. 3rd tertil), including the longitudinal experimental group classifications from Figure 1B as dependent dummy variables (each category vs. the reference normal weight group). These models also included the tanner stage and origin of children as confounding factors. When modelling the experimental groups based on obesity and IR outcomes together (classification 1), we found higher odds of being “obese or overweight with IR that remain IR after puberty entrance” in children within the 3rd tertil of the pGRS when comparing them to the children in the reference bottom pGRS group (1st and 2nd tertiles) (OR = 54.15, *p*-value = 0.008, FDR = 0.03). Higher odds of being “obese or overweight non-IR that become IR after puberty entrance” were also reported among 3rd tertile children in comparison to 1st and 2nd tertiles children though without statistical significance (OR = 15.52, *p*-value = 0.05, FDR = 0.12). Nonsignificant results were obtained for the rest of the comparisons performed. Figure 3A represents the boxplots for the continuous pGRS in each of the mentioned experimental groups. On the other hand, when modelling the experimental groups that consider only longitudinal trajectories for obesity (classification 2), we reported higher odds of being “obese remaining obese after puberty entrance” (OR = 31.91, *p*-value = 0.0009, FDR = 0.005), and “normal weight becoming overweight after puberty entrance” (OR = 26.31, *p*-value = 0.02, FDR = 0.07) among children in the 3rd tertile of the pGRS when comparing them to children in the reference bottom pGRS group (1st and 2nd tertiles). Nonsignificant results were obtained for the rest of the comparisons performed. Figure 3B represents the boxplots for the continuous pGRS in each mentioned experimental group.

### 4.3. The pGRS Does not Correlate with Increased Cardio-Metabolic Alterations in Children and Adolescents

In our cross-sectional cohort study population 1, we studied if the quartilized pGRS was associated with a metabolically unhealthy status as well as with any of its six dichotomized components (high glucose, HOMA-IR, DBP, SBP or triglycerides values or low HDLc levels) according to the criteria we have previously published [33]. In parallel, 30 continuous biochemical markers were tested for potential association with the pGRS. These biomarkers included lipid and glucose metabolism biomarkers, adipokines, as well as cardiovascular risk and proinflammatory biomarkers. From the analyses on the components of metabolic syndrome, models adjusted for BMI Z-Score, sex, age, puberty and origin showed no statistically significant association with pGRS (Appendix A). Instead, from the analyses on the 30 continuous biochemical outcomes, we found only one significant risk association for the APO B/LDLc ratio (Table 2) (that became nonsignificant after correction for multiple-hypothesis testing).

In order to validate these findings at the longitudinal level, we further applied multiple linear regressions with deltas for continuous cardio-metabolic measurements as input variables (computed as T_1_–T_0_) in the longitudinal study population 2. All analyses were again adjusted for confounders such as the change in BMI Z-score, sex, elapsed time, age at baseline or origin of the children. We found a significant positive correlation between the pGRS and APO B levels (*p*-value = 0.02, FDR = 0.29) during the course of puberty (Table 3). Moreover, a significant inverse correlation was also reported between the pGRS and the change in HDLc levels (*p*-value = 0.03, FDR = 0.33). Again, no model passed the multiple-hypothesis testing correction.

### 4.4. Lifestyle Factors Interact with the Inherited Genetic Susceptibility to Obesity in Children

Once we demonstrated the relationship between the pGRS and obesity as well as discarded a direct implication of the pGRS in the development of cardio-metabolic alterations, we next aimed to describe the plausible modulatory role of environmental factors over inherited genetic susceptibility to obesity. For that purpose, we applied multiple linear regression models including an interaction term for the pGRS and each assessed environmental factor in the study population 1. As a result, this approach revealed the pGRS to interact with three lifestyle factors related to parental educational level and physical activity (Table 4 and Figure 4). When we applied FDR adjustment for multiple testing, only two of them remained statistically significant. Interestingly, higher educational level of mothers and fathers were separately associated with lower BMI Z-Score of children depending on the pGRS (*p*-value = 0.0004 and FDR = 0.02 and *p*-value = 0.0008 and FDR = 0.02 respectively). The “protective” effect of mother’s and father’s educational levels on BMI was only achieved in children presenting low values of the pGRS (Figure 4A,B).

### 4.5. The pGRS is not Helpful for the Pharmacogenetics Management of Obesity in Children

On the other hand, we employed the data derived from a previous metformin RCT (study population 3) in order to test whether the constructed pGRS presents utility for the pharmacological management of obesity in children. As a result, we found no differential response (in terms of BMI Z-Score reduction) according to the pGRS (B = 0.39, SE = 0.41, *t* value = 0.97, *p*-value = 0.34 for the interaction term GRS*Treatment in the multiple linear regression, and *p*-value = 0.33 in for the interaction term GRS:Time:Experimental Group under a LME model).

## 5. Discussion

In the present study, we evaluated the utility of an adult-BMI pGRS for the prediction and pharmacological management of obesity in children, further investigating its implication in the appearance of cardio-metabolic alterations. For that purpose, we counted on data from three well-characterized children populations following both cross-sectional and longitudinal designs. As a result, we demonstrated that the pGRS is associated with childhood BMI Z-Score and could be used as a good predictor of obesity longitudinal trajectories during puberty. On the other hand, we demonstrated that the pGRS is not associated with cardio-metabolic comorbidities in children and that certain environmental factors interact with the genetic predisposition to the disease. Finally, according to the results derived from a weight-reduction metformin intervention in children with obesity, we discarded the utility of the pGRS as a pharmacogenetics marker of metformin response.

As one of the main findings from this work, it highlights the strongly significant association evidenced between the pGRS and the BMI Z-score in a children population composed of 698 pre- and pubertal subjects with and without obesity (Figure 1). When excluding overweight individuals, significant results were also obtained with even stronger effects sizes and a higher percent of heritability explained (Appendix A). When performing logistic regressions based on quartiles, the most significant and strongest result was obtained when comparing children in the 4th quartile of the pGRS vs. those in the reference bottom group. On the other hand, we demonstrated that only 9 over the total 44 analyzed SNPs presented an individual significant association with the BMI Z-Score, showing barely significant *p*-values (Appendix A). The *FTO* locus was identified among the most relevant loci, which is in accordance with previous studies pointing out the *FTO* as a central piece within the genetics architecture of obesity [41]. A few conclusions could be extracted from these results. The first remark is the fact that, although all assessed SNPs individually elicit small risk effects for obesity, as shown here (Appendix A) and in previous studies [3]; it is the accumulation of many of these small-risk effect variants in the same individual which finally triggers the clinical manifestation of the obesity phenotype, leading to a robust significant association (Figure 2A). This is what is known as “concerted polygenetic behavior” and has been previously described for obesity and many other complex diseases [13,42]. Under these circumstances, the use of a weighted pGRS approach is the only way to account for small risk genetics effects on disease that would otherwise remain undetected. Thus the use of pGRSs is an additional way to unravel part of the missing heritability of complex traits [13]. Furthermore, the use of a weighted approach (e.g., instead of a simple sum of the number of risk alleles by individual) improved the robustness of associations and allowed us to create a model with a higher similarity to the real to the real molecular basis of the disease.

The second remark that could be extracted from our results is the fact that the overweight status seems to be a midway phenotype (between normal weight and obesity), in which genetics might not play a determinant role (Appendix A). Although both remarks had been previously described in the literature [18,22], our approach reinforces these hypotheses and adds novel insights for Iberian populations in Spain, which is quite important considering the well-known genetic interpopulations variability within the European ancestry [43]. All these findings from our cross-sectional study populations 1 and 3 were independently validated also at each pubertal stage of the study population 2 (please see Results Section 4.2.). This corroborates the robustness of our design and reaffirms the fact that genetic predisposition to obesity starts early in childhood and persists during puberty [2]. Interestingly, the obesity heritability attributable to these genetic markers in our study was estimated in 5.6%, which is far higher than the 1–2% reported in the adult study from Locke et al. (2015) [3]. This could be explained by the fact that the environmental modulatory effects on genetics during childhood may be softer than in adults.

As a secondary aim, we demonstrated that the pGRS is useful for the prediction of obesity in children. Among all trained single-variable predictive models, the one based on the pGRS showed one of the greater predictive abilities (only surpassed by the model including parental BMI information). The joint combination of all models individually surpassing an AUC of 0.6 yielded a considerable improvement in the predictive ability (AUC = 0.81 CI 95% = [0.7–0.93]), which could be sufficient for clinical discrimination. All these results are in concordance with previous insights from Butler et al., (2019) [44], who demonstrated that early clinical factors, including maternal age, prepregnancy maternal (and paternal) BMI, birthweight, gestational age, weight gain during early infancy and other easily and measurable factors, do fairly well in predicting childhood obesity. Moreover, these results suggest that the combination of a high-risk genetic profile along with an unhealthy familial environment (represented in terms of parents BMI and obesity family history) could boost the predisposition to the disease. Beyond AUC predictive analyses, we also showed how a higher pGRS is associated with obesity longitudinal trajectories when entering puberty in study population 2. We found higher pGRS in children remaining obese after puberty when compared to children remaining with normal weight when entering puberty (Figure 3). From these results, we can conclude that the pGRS could be a powerful predictive tool, assayable from the moment of childbirth, with application in the risk assessment for future obesity. Besides, since severe obesity is usually accompanied by higher odds for metabolic complications during adulthood, these risk estimations could lead to the application of personalized preventive strategies in order to tackle the relevant problem of obesity-associated morbidity and mortality. Interestingly, as far as we are concerned, this is the first time a study focused on the longitudinal effects of a pGRS during pubertal development. Puberty has been identified as a major influence on cardiovascular risk factors, the impaired glucose tolerance of pubertal adolescents with obesity being the best explanation [24,45,46]. The demonstrated ability of the pGRS to predict, from early childhood, the pubertal obesity status of each child is therefore a tool of great interest for identifying children with higher odds for cardio-metabolic disturbances at this metabolically critical stage of life.

In previous studies performed in adults, obesity pGRSs have also yielded secondary findings for multiple cardio-metabolic outcomes, including a heightened risk of all-cause mortality, diabetes, coronary artery disease, hypertension, stroke, and venous thromboembolism, all of them after correcting for BMI. While we knew the clinical association of obesity with these outcomes and conditions, the pGRS correlation now adds the genomic underpinning. To date, no studies have demonstrated such associations in children populations otherwise. Here, we only found slightly significant associations (BMI-adjusted) between the pGRS and certain lipid metabolism outcomes (Table 2; Table 3, and Appendix A), none of them passing multiple-test adjustment. Among the rest of the analyzed outcomes, such as inflammatory and cardiovascular biomarkers, no additional significant association was found. From these results, we could conclude that the associations reported in adults between the pGRS and cardio-metabolic disturbances could be a consequence of the strong correlation between obesity, the obesogenic environment and these outcomes, rather than a direct consequence of having a higher pGRS. This is not surprising since most of the loci conforming the pGRS, 60% of them, are loci highly expressed in regions of the brain and hypothalamus regulating energy balance, appetite, food preference, and reward-seeking behavior [3], rather than loci involved in inflammatory or glucose metabolism-related processes.

From our gene-environment cross-sectional approach, we found that only the educational level of the parents demonstrated a significant interaction with the pGRS. Compared with other socioeconomic indicators, the educational level of the mother is the one that had presented the strongest association with unhealthy factors in literature, such as adiposity, in both children and adolescents [47,48]. Particularly, in our cohort, we saw how this variable was not able to break the genetic determinism or susceptibility to obesity conditioned by the pGRS. Although no other factor evidenced a genetic risk modulatory capacity in our study (neither PA measurements), this does not mean that there is no influence of the environment in the genetic predisposition to obesity in children.

Although we have previously shown that certain individual obesity-SNPs could act as pharmacogenetics regulators of metformin response in children with obesity [36], we here discarded the utility of the pGRS as a marker for the obesity pharmacological management. Again, this is not surprising given the type of SNPs included in the pGRS, where the metformin target pathways are not included. On this matter, we can conclude that a higher genetic predisposition to obesity (according to the genes involved in satiety and energy balance regulatory mechanisms) does not determine a worse BMI Z-Score response when treating with metformin. For the pharmacological personalized management of children with obesity instead, we recommend the use of individual validated target SNPs [36].

Among the limitations of our current approach, we can highlight the inclusion of only European ancestry individuals, limiting extrapolation for other ancestries and the lack of objectively measured physical activity and diet assessments. These important questions remain unanswered and will define the potential benefit derived from using this obesity pGRS.

Prevention of key medical conditions such as obesity has been a long-standing dream that largely remains unfulfilled. If we are to take advantage of the opportunity, we need to know as much as possible for prediction, acknowledging it will never be deterministic. The obesity pGRS reported in the present study provides an extremely powerful tool for the early risk detection. While there remains uncertainties and practical limitations for making such pGRS results widely available, such as the requirement for considerable education for the medical community and the general population, we are moving in the right direction for someday pre-empting important conditions that would have otherwise been manifest.

## Figures and Tables

**Figure 1 jcm-09-01705-f001:**
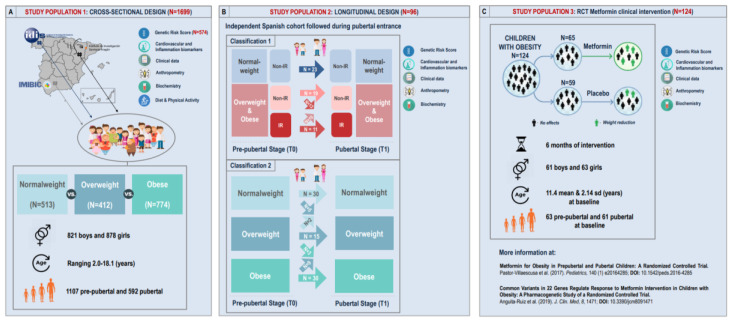
General description of study populations and design. (**A**) General characteristics of study population 1, which is based on a previously conducted case-control multicentre cross-sectional design. (**B**) General characteristics of study population 2, which is based on a previously conducted longitudinal design on 96 children undergoing puberty. (**C**) General characteristics of study population 3, which corresponds to a previous multicentre and double blind randomized controlled trial (RCT) conducted in 124 children with obesity.

**Figure 2 jcm-09-01705-f002:**
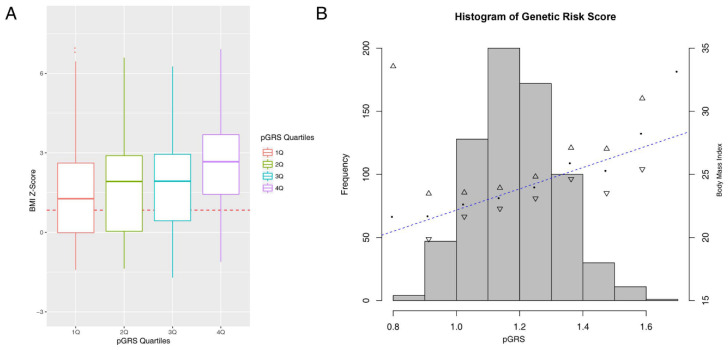
Association between polygenetic risk scores (pGRS) and body mass index (BMI) Z-score in the study population 1; analysis adjusted for origin and pubertal status of children. (**A**) Boxplot graph for BMI Z-Score according to each quartile of the pGRS; the dashed line in the plot represents the cut-off BMI Z-Score for overweight and obesity in the study population 1. (**B**) Histogram of genetic risk score values in the study population 1 and their correlation with BMI (R^2^ = 0.2).

**Figure 3 jcm-09-01705-f003:**
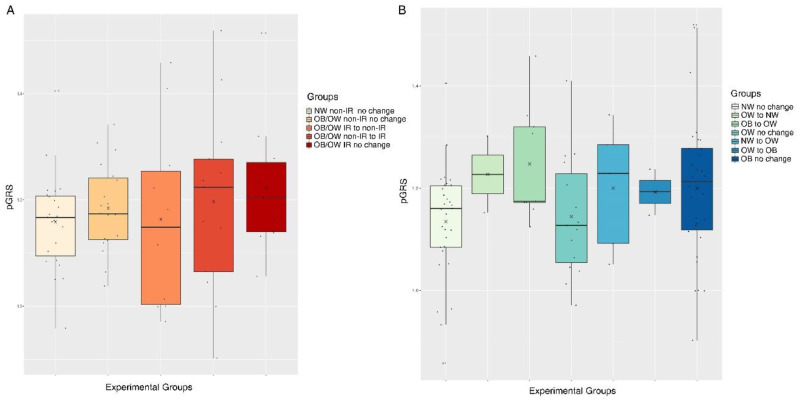
Boxplots for the continuous pGRS according to the two available experimental group classifications of study population 2. (**A**) pGRS boxplots according to joint longitudinal trajectories for obesity and insulin resistance (IR). (**B**) pGRS boxplots according to the longitudinal trajectories for obesity. The x symbol in plots represents the mean pGRS for each group. Abbreviations: NW, normal weight; OB, obese; OW, overweight; IR, insulin resistance.

**Figure 4 jcm-09-01705-f004:**
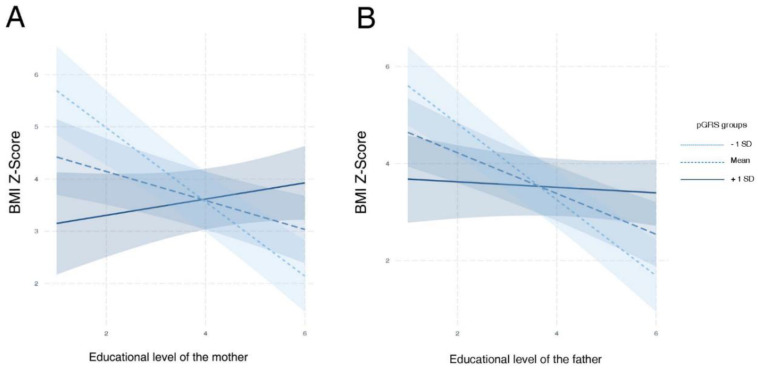
Interaction plots for observed significant modulatory effects of environment over inherited genetic susceptibility to obesity (pGRS-environment interactions) in study population 1; these analyses were adjusted for the origin and pubertal status of children. (**A**) shows the modulatory effect of the educational level of mothers over the pGRS-BMI Z-Score association. (**B**) shows the modulatory effect of the educational level of fathers over the pGRS-BMI Z-Score association. In both subfigures, the pGRS is categorized according to the cut off values −1 standard deviation and +1 standard deviation. Abbreviations: BMI, body mass index; SD, standard deviation.

**Table 1 jcm-09-01705-t001:** Obesity predictive ability for assessed traditional and genetics risk factors in the study population 1.

		Whole Population			Training Set			Test Set		
Predictors	AUC [95% CI]	n	n Normal-Weight	n Obese	n	n Normal-Weight	n Obese	n	n Normal-Weight	n Obese
Tanner, Origin, Sex and Age	0.66 [0.61–0.72]	1285	512	773	901	359	542	384	153	231
pGRS	0.72 [0.63–0.80]	443	187	256	311	131	180	132	56	76
Obesity Family History	0.70 [0.63–0.77]	686	232	454	481	163	318	205	69	136
Maternal Smoking	0.49 [0.43–0.55]	632	218	414	443	153	290	189	65	124
Gestational Diabetes	0.49 [0.45–0.54]	620	214	406	435	150	285	185	64	121
Birthweight	0.60 [0.51–0.69]	610	211	399	428	148	280	182	63	119
Gestational Weight Gain	0.54 [0.45–0.62]	569	206	363	400	145	255	169	61	108
Parents BMI	0.76 [0.68–0.84]	530	199	331	372	140	232	158	59	99
Type of Breastfeeding	0.55 [0.45–0.64 ]	555	193	362	390	136	254	165	57	108
Tanner, Origin, Sex, Age, pGRS, Obesity Family History, Birthweight, and Parents BMI	0.81 [0.7–0.93]	176	78	98	124	55	69	52	23	29

Models were first constructed based on each risk factor alone and then combined according to the improvements in area under the curve (AUC). For this purpose, subjects with normal weight and with obesity were selected from the study population 1 and randomly assigned to training and test subsets in whichpredictive models were trained and evaluated, respectively. Individual models showing an AUC ≥ 0.60 were combined into the full model presented in the last row. Abbreviations: AUC, area under the curve; BMI, body mass index; CI, confidence interval; n; number of effective individuals for analysis; pGRS, polygenetic risk score.

**Table 2 jcm-09-01705-t002:** Association between the pGRS and metabolic outcomes in the cross-sectional cohort of 574 children (study population 1), in decreasing order of statistical significance.

Measurement	Beta	SE	CI.LO	CI.HI	T-Value	*p*-Value	FDR
APO B/LDLc Ratio	−0.12	0.05	−0.21	−0.02	−2.29	0.02	0.60
Triglycerides (mg/dL)	−17.69	9.17	−35.67	0.28	−1.93	0.05	0.68
APO B (mg/dL)	−10.94	6.49	−23.66	1.78	−1.69	0.09	0.68
APO A/APO B	0.40	0.25	−0.09	0.88	1.61	0.11	0.68
WC/Height Ratio	0.03	0.02	−0.01	0.06	1.51	0.13	0.68
WC (cm)	3.75	2.52	-1.19	8.70	1.49	0.14	0.68
Adiponectin/Leptin Ratio	0.50	0.36	-0.21	1.22	1.38	0.17	0.68
MCP1 (ng/L)	−21.94	16.22	−53.73	9.84	−1.35	0.18	0.68
aPAI (ug/L)	−3.56	2.90	−9.24	2.12	−1.23	0.22	0.69
IL8 (ng/L)	−0.60	0.54	−1.66	0.45	−1.12	0.26	0.69
QUICKI	0.01	0.01	−0.01	0.03	0.96	0.34	0.69
DBP (mmHg)	3.11	3.35	−3.46	9.68	0.93	0.35	0.69
Adiponectin (mg/L)	−3.09	3.34	−9.63	3.45	−0.93	0.35	0.69
IL6 (ng/L)	2.32	2.57	−2.71	7.35	0.90	0.37	0.69
HC (cm)	1.96	2.19	−2.32	6.25	0.90	0.37	0.69
HOMA-IR index	−0.37	0.42	−1.19	0.46	−0.87	0.38	0.69
WC/HC Ratio	0.02	0.03	−0.03	0.08	0.85	0.39	0.69
Total cholesterol (mg/dL)	−5.95	9.48	−24.53	12.63	−0.63	0.53	0.88
hsCRP (mg/L)	0.42	0.81	−1.16	2.01	0.52	0.60	0.89
HDLc/LDLc Ratio	−0.05	0.09	−0.22	0.13	−0.49	0.62	0.89
Glucose (mg/dL)	−1.14	2.38	−5.81	3.53	−0.48	0.63	0.89
Resistin (ug/L)	1.28	2.88	−4.37	6.93	0.44	0.66	0.89
SBP (mmHg)	−1.74	4.17	−9.92	6.44	−0.42	0.68	0.89
LDLc (mg/dL)	−2.43	8.57	−19.21	14.36	−0.28	0.78	0.89
TNF (ng/L)	0.15	0.55	−0.93	1.24	0.28	0.78	0.89
Leptin (ug/L)	0.77	2.86	−4.84	6.37	0.27	0.79	0.89
APO A (mg/dL)	−2.35	9.74	−21.44	16.75	−0.24	0.81	0.89
MPO (ug/L)	−2.17	10.09	−21.95	17.60	−0.22	0.83	0.89
HDLc (mg/dL)	−0.50	4.22	−8.76	7.77	−0.12	0.91	0.94
MMP9 (ug/L)	−0.05	20.57	−40.38	40.27	0.00	1.00	1.00

Multiple linear regression analyses with the pGRS as independent variable were run adjusted for sex, BMI Z-Score, origin and puberty. When the dependent variable was the blood pressure, we further added the height as a confounder in the model. Abbreviations: APO, apolipoprotein; CI.HI, high confidence interval; CI.LO, low confidence interval; DBP, diastolic blood pressure; FDR, false discovery rate; HC, hip circumference; HDLc, high-density lipoproteins-cholesterol; HOMA-IR, homeostasis model assessment for insulin resistance; hsCRP, high-sensitivity C reactive protein; IL, interleukin; LDLc, low-density lipoproteins-cholesterol; MCP1, monocyte chemoattractant protein 1; MMP9, Matrix metallopeptidase 9; MPO, myeloperoxidase; PAI-1, plasminogen activator inhibitor-1; QUICKI, quantitative insulin sensitivity check index; SBP, systolic blood pressure; SE, standard error; TNF-α, tumour necrosis factor alpha; WC, waist circumference.

**Table 3 jcm-09-01705-t003:** Association between the pGRS and deltas for continuous cardio-metabolic measurements (computed as T_1_–T_0_) in the longitudinal study population 2.

Measurement (Delta T_1_–T_0_)	Beta	SE	CI.LO	CI.HI	T-Value	*p*-Value	FDR
APO B (mg/dL)	57.44	21.63	15.05	99.82	2.66	0.02	0.29
HDLc (mg/dL)	−18.91	8.98	−36.51	−1.32	−2.11	0.03	0.33
Triglycerides (mg/dL)	45.38	31.58	−16.51	107.27	1.44	0.15	0.78
APO A (mg/dL)	−35.22	30.97	−95.92	25.48	−1.14	0.26	0.78
DBP (mmHg)	−13.34	12.98	−38.79	12.10	−1.03	0.31	0.78
Insulin (mU/L)	8.94	8.89	−8.49	26.37	1.00	0.32	0.78
SBP (mmHg)	15.57	16.77	−17.29	48.43	0.93	0.36	0.78
HOMA-IR index	1.88	2.07	−2.17	5.94	0.91	0.37	0.78
HDLc/LDLc Ratio	−0.29	0.43	−1.14	0.56	−0.67	0.51	0.86
Total cholesterol (mg/dL)	−13.63	24.38	−61.42	34.16	−0.56	0.58	0.86
Glucose (mg/dL)	−3.80	8.72	−20.89	13.29	−0.44	0.66	0.86
WC/HC	−0.03	0.08	−0.19	0.12	−0.41	0.68	0.86
WC (cm)	−3.11	9.17	−21.09	14.87	−0.34	0.74	0.86
LDLc (mg/dL)	−5.94	18.41	−42.02	30.15	−0.32	0.75	0.86
QUICKI	0.01	0.04	−0.06	0.09	0.31	0.75	0.86

Multiple linear regression analyses with the pGRS as independent variable were run adjusted for sex, the change in BMI Z-Score, the origin, the elapsed time from baseline to puberty as well as the pubertal stage reached. When the dependent variable was the change in blood pressure, we further added the change in height as a confounder in the model. Abbreviations: APO, apolipoprotein; CI.HI, high confidence interval; CI.LO, low confidence interval; DBP, diastolic blood pressure; FDR, false discovery rate; HC, hip circumference; HDLc, high-density lipoproteins-cholesterol; HOMA-IR, homeostasis model assessment for insulin resistance; hsCRP, high-sensitivity C reactive protein; LDLc, low-density lipoproteins-cholesterol; QUICKI, quantitative insulin sensitivity check index; SBP, systolic blood pressure; SE, standard error; WC, waist circumference.

**Table 4 jcm-09-01705-t004:** Interaction analyses between the pGRS and each assessed environmental factor in the study population 1.

Lifestyle Factor	Beta	SE	CI.LO	CI.HI	T-Value	*p*-Value	FDR
Educational level of the mother	3.41	0.95	1.56	5.27	3.60	3.86 × 10^−4^	0.02
Educational level of the father	2.93	0.86	1.24	4.62	3.40	7.92 × 10^−4^	0.02
How many minutes per week do you spend exercising at a sport program?	0.02	0.01	0.00	0.04	2.13	0.03	0.47
Presence of AH in father or mother	−4.01	2.01	−7.95	−0.08	−2.00	0.05	0.56
Mother BMI	−0.22	0.11	−0.44	0.00	−1.93	0.06	0.56
How long does it take to get to the school on walk?	−0.24	0.14	−0.52	0.04	−1.66	0.10	0.58
How often do you eat fruit while watching TV?	1.73	1.05	−0.34	3.80	1.64	0.10	0.58
How often do you eat snacks while watching TV?	1.80	1.14	−0.44	4.03	1.58	0.12	0.58
How many hours do you spend doing home activities?	−2.55	1.62	−5.73	0.63	−1.57	0.12	0.58
How much time do you play videogames in a day during weekend?	1.03	0.70	−0.35	2.41	1.46	0.15	0.58
How many hours each day do you spend doing vigorous efforts like training activity?	2.27	1.55	−0.77	5.32	1.46	0.15	0.58
Presence of hypercholesterolemia in father or mother	1.78	1.24	−0.64	4.21	1.44	0.15	0.58
Presence of heart stroke in father or mother	−9.74	6.86	−23.19	3.70	−1.42	0.16	0.58
Presence of vascular problems in father or mother	−39.77	29.84	−98.26	18.72	−1.33	0.18	0.59
How many hours do you spend exercising in a sport club?	0.04	0.03	−0.02	0.10	1.30	0.19	0.59
How often do you eat salted potatoes while watching TV?	2.11	1.65	−1.12	5.34	1.28	0.20	0.59
Presence of diabetes in father or mother	11.59	9.96	−7.94	31.12	1.16	0.25	0.64
How often do you eat nuts while watching TV?	2.21	1.95	−1.62	6.04	1.13	0.26	0.64
How many days per week do you spend walking with vigorous efforts?	0.63	0.56	−0.47	1.73	1.13	0.26	0.64
How many hours each day do you spend practicing activities that do not require physical activity (e.g., reading)	0.41	0.44	−0.44	1.27	0.95	0.34	0.77
How much time do you play videogames in a day during the week?	−0.79	0.85	−2.46	0.87	−0.93	0.35	0.77
How many hours do you usually sleep every day during the week?	0.68	0.80	−0.89	2.26	0.85	0.40	0.77
How many hours do you spend doing physical activity in family?	1.21	1.42	−1.58	4.00	0.85	0.40	0.77
How often do you eat candies while watching TV?	1.50	1.77	−1.96	4.96	0.85	0.40	0.77
How many hours do you usually sleep every day during the weekends?	−0.55	0.67	−1.85	0.76	−0.82	0.41	0.77
Diagnosed hypertriglyceridemia in father or mother	1.06	1.34	−1.57	3.69	0.79	0.43	0.77
How often do you eat sweets while watching TV?	1.37	1.78	−2.12	4.86	0.77	0.44	0.77
How much time per weekend do you usually use the smartphone?	0.69	0.95	−1.17	2.55	0.73	0.47	0.79
Do you usually eat in front of the TV?	0.77	1.15	−1.49	3.02	0.67	0.51	0.83
How many hours per week do you spend on physical education during school hours?	−2.01	3.48	−8.83	4.82	−0.58	0.57	0.85
How often do you eat fruits while playing video games?	−2.07	3.73	−9.38	5.23	−0.56	0.58	0.85
Presence of obesity in the father or mother	−0.45	0.82	−2.07	1.17	−0.54	0.59	0.85
How much time during the week do you usually watch TV?	0.39	0.75	−1.07	1.85	0.52	0.60	0.85
How often do you eat fruits while surfing internet?	1.81	3.88	−5.80	9.43	0.47	0.64	0.88
How much time in per weekend do you usually use internet	0.25	0.63	−0.98	1.48	0.40	0.69	0.92
How many hours a day do you spend walking with vigorous efforts?	−0.35	1.00	−2.32	1.62	−0.35	0.73	0.92
How often do you eat snacks while surfing the internet?	1.26	3.77	−6.13	8.65	0.33	0.74	0.92
How many days per week do you spend doing home activities?	0.20	0.64	−1.05	1.44	0.31	0.75	0.92
How many days per week do you spend doing physical activity in family?	−0.26	0.91	−2.05	1.53	−0.29	0.78	0.92
How many hours each day you spend walking quite a lot without vigorous efforts?	0.17	0.63	−1.07	1.40	0.26	0.79	0.92
Father BMI	−0.04	0.17	−0.38	0.30	−0.25	0.80	0.92
How often do you eat snacks while playing videogames?	−0.73	3.99	−8.56	7.09	−0.18	0.85	0.94
How much time during the weekend do you spend watching TV and DVD?	0.14	0.79	−1.41	1.70	0.18	0.86	0.94
How many days per week do you exercise in a sport club?	0.07	0.69	−1.28	1.43	0.11	0.92	0.98
How many hours do you spend doing homework outside of school hours?	0.05	0.99	−1.88	1.99	0.05	0.96	0.99
How much time per day do you use the internet during the week?	−0.02	0.71	−1.41	1.37	−0.03	0.98	0.99
How many days do you spend doing vigorous efforts like training activity?	−0.01	0.62	−1.22	1.19	−0.02	0.99	0.99

These analyses were adjusted for origin and pubertal status of children. Abbreviations: AH, arterial hypertension; BMI, body mass index; CI.HI, high confidence interval; CI.LO, low confidence interval; FDR, false discovery rate; SE, standard error.

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
