# Peer review of "Evaluation of the Predictive Ability, Environmental Regulation and Pharmacogenetics Utility of a BMI-Predisposing Genetic Risk Score during Childhood and Puberty"

_jcm, 2020, doi:10.3390/jcm9061705_

Round 1
Reviewer 1 Report
In the current manuscript., the authors investigated the relationship between pGRS and obesity during the course of childhood and puberty. The major findings demonstrated BMI pGRS could be used to predict the children obesity before and after the onset of puberty. Overall, the experimental study is well designed and the results support their conclusions.
- Did the authors ever consider sleeping during could be associated with the genetic susceptibility to obesity?
Minor points
- Please use bigger fonts for the labels in all the figures in order to make them visible.
Reviewer 2 Report
In this study, authors documented the association between a polygenic risk score, which include 56 BMI-associated SNPs, with obesity, numerous cardiometabolic variables, change in obesity status during puberty, pharmalogical management of obesity, and the answer obtained to 47 lifestyle questions. This study addresses an interesting question and the manuscript could have so been interesting if it would have not been so long and confusing. Unfortunately, it seems that authors were unable to identify the most important variables, the most interesting analyses to perform and the most valuable results to answer their objectives.
Why the authors did not choose to construct the pGRS based on the most determinant SNPs?
Section 4.1. Although strongly significant, the correlation is not strong (it would be interesting for readers to see the correlation coefficient). Besides, according to the Figures, it appears to have an important overlap of pGRS between normal-weight and obese subjects.
Table 1. To perform 17 logistic regression models to find the best one looks like a fishing expedition. What was the objective behind this strategy? Most of these models are not statistically different.
Authors have performed logistic analyses to document the ability of pGRS to predict future outcomes after puberty. To do so, they have used pGRS as a dependant variable. Predictor variables must always be considered as independent variables.
